# JNK Signaling as a Key Modulator of Soft Connective Tissue Physiology, Pathology, and Healing

**DOI:** 10.3390/ijms21031015

**Published:** 2020-02-04

**Authors:** Georgia Nikoloudaki, Sarah Brooks, Alexander P. Peidl, Dylan Tinney, Douglas W. Hamilton

**Affiliations:** 1Department of Anatomy & Cell Biology, Schulich School of Medicine and Dentistry, University of Western Ontario, 1151 Richmond St, London, ON N6A 5C1, Canada; gnikolou@uwo.ca; 2Biomedical Engineering Graduate Program, Schulich School of Medicine and Dentistry, University of Western Ontario, 1151 Richmond St, London, ON N6A 5C1, Canada; sbrook27@uwo.ca (S.B.); dtinneyd@uwo.ca (D.T.); 3Department of Physiology and Pharmacology, Schulich School of Medicine and Dentistry, University of Western Ontario, 1151 Richmond St, London, ON N6A 5C1, Canada; apeidl@uwo.ca; 4Division of Oral Biology, Schulich School of Medicine and Dentistry, University of Western Ontario, 1151 Richmond St, London, ON N6A 5C1, Canada

**Keywords:** wound healing, skin, cornea, tendon, gingival tissue, dental pulp, imaginal discs

## Abstract

In healthy individuals, the healing of soft tissues such as skin after pathological insult or post injury follows a relatively predictable and defined series of cell and molecular processes to restore tissue architecture and function(s). Healing progresses through the phases of hemostasis, inflammation, proliferation, remodeling, and concomitant with re-epithelialization restores barrier function. Soft tissue healing is achieved through the spatiotemporal interplay of multiple different cell types including neutrophils, monocytes/macrophages, fibroblasts, endothelial cells/pericytes, and keratinocytes. Expressed in most cell types, c-Jun N-terminal kinases (JNK) are signaling molecules associated with the regulation of several cellular processes involved in soft tissue wound healing and in response to cellular stress. A member of the mitogen-activated protein kinase family (MAPK), JNKs have been implicated in the regulation of inflammatory cell phenotype, as well as fibroblast, stem/progenitor cell, and epithelial cell biology. In this review, we discuss our understanding of JNKs in the regulation of cell behaviors related to tissue injury, pathology, and wound healing of soft tissues. Using models as diverse as *Drosophila*, mice, rats, as well as human tissues, research is now defining important, but sometimes conflicting roles for JNKs in the regulation of multiple molecular processes in multiple different cell types central to wound healing processes. In this review, we focus specifically on the role of JNKs in the regulation of cell behavior in the healing of skin, cornea, tendon, gingiva, and dental pulp tissues. We conclude that while parallels can be drawn between some JNK activities and the control of cell behavior in healing, the roles of JNK can also be very specific modes of action depending on the tissue and the phase of healing.

## 1. Introduction

In soft connective tissues, the response to pathological insult or mechanical injury is a well-coordinated spatiotemporal set of cell and molecular processes to restore tissue integrity and function [1,2]. Multiple changes occur in both the physical and chemical signals present in the tissue microenvironment as a result of damage or pathology [3]. A plethora of cell types including inflammatory cells, mesenchymal cells, pericytes, endothelial cells, and epithelial cells are recruited to the site of injury in a coordinated and temporal fashion [4]. Each distinct cell population assumes a specific phenotype in healing, which is in part regulated through these altered cues present in the microenvironment created in response to injury or pathology. In order to promote cell behaviors required for healing, it is necessary for alterations in extracellular stimuli to be recognized and translated into molecular signals that regulate processes including adhesion, migration, proliferation, apoptosis, gene expression, and matrix synthesis. Linking the extra- and intracellular compartments, there are numerous intracellular signaling pathways which essentially translate extracellular signals into changes in cell phenotype. 

c-Jun N-terminal kinases (JNKs), which are members of the mitogen-activated protein kinase (MAPK) family, are known to be important molecules that mediate the intracellular response of eukaryotic cells to many different types of stimuli present in the external cellular microenvironment [5]. JNKs have also been shown to be activated in response to stress [6], cytokines [7], growth factors [8,9,10,11,12,13,14], extracellular matrix molecules [15], as well as pathogens [16], all of which are present after tissue injury. JNKs are expressed in all cell types, with 10 isoforms identified that arise from the alternate splicing of three distinct genes [17]. JNK-1 and JNK-2 are expressed throughout the body, with only JNK-3 showing more restricted expression specifically in brain, heart, and testis [18]. 

The focus of this review is to discuss the roles of JNKs in cellular processes essential in the injury response, pathology, and healing process of soft connective tissues. In this article we focus on the role of JNKs specifically in the healing of *Drosophila* imaginal discs, skin, tendon, cornea, and the focus of our laboratory, oral tissues (gingiva and dental pulp). Although the role of JNKs have been well described in many soft tissues including skin, as well as in pathological conditions including cancer [19], diabetes [20], and neurodegeneration [21], its role in normal soft tissue healing processes is only now being elucidated, however, conflicting findings have arisen concerning the role of JNKs. Traditionally, considered a response to cellular stress, tissue healing has demonstrated a versatile role for JNK signaling in response to injury in several different tissues. 

## 2. Drosophila Melanogaster as a Model to Study JNK in Wound Repair and Re-Epithelialization

Since the early 1900s, *Drosophila melanogaster* has been used extensively in research [22]. Due to many unique characteristics, including short life cycles, good durability, and easy genetic manipulability, *Drosophila* are still widely used in many aspects of genetic and physiological research [22]. Genetic manipulation of *Drosophila* can provide powerful tools to study specific signaling pathways in vivo [23]. The process of wound healing is a well-conserved physiological response in *Drosophila* that also shares many aspects with processes evident in mammalian wound healing [24,25]. Studies using *Drosophila* model systems have revealed important roles for JNKs in wound repair. Below, we summarize some of these recent findings, and highlight the importance of *Drosophila* as a genetic model for studying JNK signaling in wound repair. 

Wound closure in *Drosophila* requires directional migration of an epithelial sheet towards the center of the wound, a process known as re-epithelialization. Similar to the mammalian wound healing process, cells must polarize, change shape, and coordinate cell–cell and cell–matrix interactions [26,27]. Using the pinch wound model of wound repair in *Drosophila,* it has been shown that these cellular processes of wound healing are in part regulated by JNK pathways in *Drosophila*. Specifically, JNK plays a role in myosin II localization and function in cells of the migrating epithelial sheet [28]. This localization of myosin requires expression and activity of specific integrin subunits, downstream of JNK activation [29]. Loss of function studies have shown that Rac1, Rho1, and Cdc42 are all required for coordinated polarization of cells at the leading edge of the wound. These three GTPases are upstream regulators of JNK and appear to function in a partially redundant manner [30]. In a similar model of wound healing, it has also been shown that loss of function of the JNK signaling factors Slipper (*Slpr*), Hemipterous (*Hep*), *D-fos,* and *D-jun* results in wound closure defects [31]. Using tissue-specific gene expression of transgenes, loss-of-function studies have also identified the transcriptional coactivator Yorkie (*Yki*) as an important player in wound closure. *Yki* has been shown to interact with members of the JNK pathway during healing and interference with *Yki* function results in impaired wound closure. It is thought that *Yki* plays a role in effective actin cable formation during wound closure [32]. Furthermore, the role of JNK in re-epithelialization in *Drosophila* can act downstream of Cdc37 activation; *Drosophila* lacking Cdc37 exhibits reduced JNK signaling and impaired healing [33]. In addition to these important roles in cell polarization and migration, JNK has also been shown to play an important role in cell fusion during *Drosophila* re-epithelization. In *Drosophila*, JNK appears to regulate spatiotemporal cell fusion during wound repair in a coordination with JAK/STAT signaling [33]. 

Another useful research model of wound repair and regeneration in *Drosophila* involves tissue formation through a proliferative repair process [34]. As with other models of wound repair in *Drosophila*, JNK plays an important role in imaginal disc regeneration. Components of the JNK pathway have been shown to play roles in migration and the spread of the leading edge of cells surrounding the wound area, specifically having effects on actin cable and filopodia formation [34]. Other uses of imaginal discs, in the study of wound repair, involve an intricate model of cell death-induced regeneration. In a recent study, restricted induction of cell death was used to activate a wound repair process and found that the JNK pathway is activated at the leading edges of the healing tissue in this model, and plays a role in early regeneration events, likely though induction of proliferation [35].

During the developmental process of dorsal closure in *Drosophila*, epithelial movement resembles morphological aspects of mammalian wound repair. The dorsal-most cells of the lateral epidermis form the leading edge of an epithelial sheet. The cells elongate along the dorsal axis and the epithelial sheet moves dorsally until closure is complete [36]. Many of these genes involved in this process are also conserved in mammals. Using mutant strains of *Drosophila*, it has been shown that JNK signaling is critical in dorsal closure; *puckered (puc)*, a critical downstream target of JNK in *Drosophila*, is induced at the edge of the wound and can be critical for induction of shape change and migration [37]. The use of the *Drosophila* dorsal closure model to study JNK signaling has been well documented in a previous review article [38].

Taken together, these studies highlight the importance of JNK signaling in *Drosophila* wound healing. Due to the evolutionarily conserved nature of the process of wound healing and regeneration, many of the results obtained from research in *Drosophila* can be used to further our understanding of mammalian wound healing. It is important to appreciate the power of *Drosophila* in research, especially when studying the complex process of wound healing. 

## 3. Skin and Wound Repair

As is evident from studies in *Drosophila*, JNK plays important roles in the regulation of epithelial migration and proliferation, processes which are also essential in mammalian skin healing. As a tissue, skin has been a prominent model used to investigate inflammation, as well as fibroblast and keratinocyte biology in healing, despite known differences between mice and humans in skin structure and physiology [39,40,41,42,43,44,45]. The pattern of JNK proteins and their activation in mammalian skin healing have been shown to vary both with time and cell type. Xiong and colleagues investigated when JNK levels peak during incisional skin healing in mice, with a detectable phospho-JNK increase in neutrophils as early as three hours after injury [46]. Between day one and five, phospho-JNK was mainly localized to mononuclear cells and fibroblasts and by days seven to 14, mainly in mesenchymal cell populations. Because there is a well described transition in cell populations during healing from neutrophils (early inflammation) to monocytes and macrophages (mid to late inflammation) to mesenchymal cells (proliferative and remodeling phases), these findings suggest that most cell types involved in the wound healing processes activate JNK signaling at each stage, which is discussed below. 

### 3.1. Inflammation

With specific emphasis on inflammatory cell populations, Xiong et al. showed that the ratio of phospho-JNK positive cells to total cell number increased gradually in incisional wounds from three hours to one day post wounding, with maximal positivity of phospho-JNK present 24 hours following injury [46], strongly implicating JNK signaling in neutrophils’ behavior which are the predominant cell population in the wounds at this timepoint. It is known that pro-inflammatory growth factor tumor necrosis factor α (TNF-α), stimulates activation of both JNK-1 and JNK-2 isoforms in human neutrophils [47]. Similar to staining patterns evident in incisional wound healing, JNK levels have also been shown to peak at day one following burn injury in a murine model [48]. In normal skin, the positive signals for phospho-JNK were predominantly localized to the basal layer cells of the epidermis. However, in the experimental burn group, phospho-JNK localized to epidermal cells and polymorphonuclear cells, with a significantly higher immunoreactivity for JNK evident in comparison with control groups [48]. 

Therefore, it appears that any injury to the skin, whether due to incisional wounding or burn injury, results in a peak of JNK activation at 24 hours post injury. Interestingly, it is known that TNF-α mediated JNK phosphorylation in neutrophils leads to rapid apoptosis [49], but only through integrin mediated neutrophil adherence; neutrophils in suspension are unaffected. Neutrophils themselves are a significant producer of pro-inflammatory cytokines including TNF-α, following injury [50]. The possibility exists that JNK activation in neutrophils caused by TNF-α, stimulating apoptosis, could represent a feedback loop to begin the transition of the wound towards an anti-inflammatory phenotype 24 hours post wounding (Figure 1). Further evidence for this hypothesis is highlighted by the observation that JNK phosphorylation does not actually play any role in the regulation and production of inflammatory cytokines by neutrophils [51]. In contrast to neutrophils, in monocyte and macrophage populations, which begin to infiltrate granulation tissue at 24 hours following injury, JNK signaling has been shown to regulate their survival [52,53]. Therefore, neutrophil and macrophage populations activate JNK signaling, although it could mediate apoptosis of one population and survival of the other. However, other studies have shown that TNF-α can induce apoptosis of macrophages [54], suggesting that significant crosstalk between JNKs and other signaling cascades likely determines the fate of cells in a given microenvironment. While TNF-α is associated mainly with inflammatory cells and processes, it also plays a role in dermal fibroblast activation post wounding, although conflicting data has emerged with respect to JNK roles in fibroblast biology, as is highlighted below.

### 3.2. Fibroblast Behavior

Post injury to the skin, dermal fibroblasts and other mesenchymal populations are recruited to the developing granulation tissue in the wound bed where they assume a matrix secreting, contractile phenotype known as a myofibroblast [42]. As highlighted in Section 3.1, during the inflammatory phase of healing, TNF-α is secreted first by neutrophils and, subsequently, monocytes and macrophages and it has been shown that TNF-α activates JNK signaling in dermal fibroblasts [55]. While certainly not conclusive, JNK knockout fibroblasts show a reduced migration ability, which is replicated in human dermal fibroblasts by pharmacological inhibition of JNK signaling [56]. This suggests that the TNF-α containing microenvironment present in the inflammatory phase of healing can actually promote fibroblast recruitment into the granulation tissue through JNK activation (Figure 1), although TGFβ is currently considered the predominant cytokine to stimulate this invasion of fibroblasts [57]. However, TNF-α has been shown to be chemotactic for dermal fibroblasts [58], suggesting there could be redundancy present in the system to ensure fibroblast are successfully recruited to granulation tissue. Indeed, the addition of anti-TNF-α antibodies reduces fibroblast (and inflammatory cell) density in skin wounds at day three post wounding [59]. The transition of the inflammatory phase to the proliferative phase of healing is associated with reduced levels of TNF-α and increasing levels of TGFβ [60]. Of significance, TNF-α and TGFβ are antagonistic [61], with TNF-α actually inhibiting Smad3 mediated transcription through activation of c-Jun and JunB [62].

Upon colonization of the granulation tissue, dermal fibroblasts transition to the myofibroblast phenotype, which is associated with α-smooth muscle actin (α-SMA) expression, extracellular matrix production, and wound contraction [42,43,45]. Mediated by TGFβ, collagen production is associated with JNK activation resulting in phosphorylation of its downstream target c-Jun. Inhibition of c-Jun phosphorylation prevents TGFβ mediated collagen production [63]. TGFβ induction of the expression of profibrotic genes, such as collagen, is dependent on upregulation of endothelin-1 (ET-1) [63] (Figure 2). Of importance, TGFβ induces endothelin-1 (ET-1) expression in human dermal fibroblasts in a manner dependent on Smad and activator protein 1 and JNK-dependent signaling [63]. Further evidence of the importance of JNK signaling in extracellular matrix production by fibroblasts has been provided by skin fibrosis models [64]. Inhibition of JNK significantly reduced dermal thickening, myofibroblast differentiation, and collagen deposition in a dose-dependent manner in mice [64]. With respect to cellular contraction, addition of JNK inhibitors significantly reduced connective tissue growth factor (CCN2)-induced expression of α-SMA and collagen type I in human fibroblasts isolated from hypertrophic scars [65]. Furthermore, using a rabbit ear model of the hypertrophic scarring, the authors demonstrated that addition of JNK inhibitors significantly reduced scar formation [65]. Previous studies have shown that mouse embryonic fibroblasts overexpressing CCN2 show increased phosphorylation of JNK in comparison to wild-type cells, which was concomitant with increased production of fibronectin, collagen, and α-SMA [66]. 

Taken together, these findings show that inhibition of JNK signaling appears to reduce matrix production and contraction. On the one hand, contrasting results are present in the literature. Dolivo and colleagues demonstrated using human dermal fibroblasts that inhibition of JNK actually activated fibroblasts [67]. Quantification revealed increased expression of α-SMA, calponin 1, the mRNA transcripts *Col1A1, Col1A2, Col3A1*, and the myofibroblast-specific fibronectin isoform (*ED-A Fn)* (Figure 2). On the other hand, other studies have shown similar findings. JNK knockout fibroblasts have shown increased contraction as compared with wild-type fibroblasts but did show similar transcript levels for collagen [56]. In summary, the current literature is conflicting with respect to the role of JNK signaling in fibroblast behavior and further work is required. 

### 3.3. Keratinocyte Behavior

Concomitant with inflammatory and proliferative processes post injury, to restore barrier function, keratinocyte migration is initiated. In molecular terms, keratinocytes have to switch gene expression patterns from that associated with differentiation (markers include cytokeratin 14, involucrin, and filaggrin [68]) to one required for migration. Wound edge keratinocytes downregulate keratins to facilitate the proliferation and migration process [69]. JNK signaling is strongly implicated in the regulation of these processes, although conflicting data exists concerning the role of JNKs specifically. With respect to keratinocyte differentiation, Koehler and colleagues demonstrated that mice deficient in JNK1, but not JNK2 or JNK3, exhibited a significant delay in re-establishment of the permeability barrier repair after superficial injury to the surface of the skin, as well as a delay in the healing of full thickness excisional wounds [70]. Secondarily, they showed that JNK1 activity increased four-fold and strongly correlated with the degree of differentiation in organotypic keratinocyte cultures, suggesting that JNK-1 is required for keratinocyte stratification and differentiation [70]. These studies were supported by research that showed that the inhibition of JNK in normal human epidermal keratinocytes suppressed expression of filaggrin, a protein essential for maintenance of barrier function [71]. In contrast, Gazel and colleagues demonstrated that inhibition of JNK in epidermal keratinocytes initiated their differentiation program [72]. Specifically, they quantified the transcriptional profiles of human neonatal foreskin keratinocytes and SP600125 treated keratinocytes, which inhibited JNK signaling [10,73]. They identified that inhibition of JNK resulted in suppression of cell division and motility, upregulation of differentiation markers, and formation of cornified envelopes as compared with untreated control cultures [72] (Figure 3). 

Activity of MMP-1 is required for human keratinocytes to migrate on collagen, although inhibiting JNK signaling has no effect on MMP-1 production [74]. Further evidence of the lack of JNK involvement in keratinocyte migration was provided by Li and colleagues who demonstrated that human keratinocyte motility in response to collagen pre-coated plates did not require JNK activity [75]. Moreover, p38MAP kinase have been shown to regulate many aspects of keratinocyte migration [76], although in response to certain molecules including activin B, RhoA-Rock-JNK-c-Jun involvement is implicated [77]. With the rising incidences of non-healing skin ulcers, understanding the exact role of JNK signaling and, in particular, the individual isoforms, in keratinocyte migration and differentiation, could highlight potential new therapeutics to stimulate closure in impaired healing. 

## 4. Corneal Injury and Healing

The cornea, the protective outer layer of the eye, is one of the most common sites of trauma and inflammation in the optical system due to its superficial location [78]. Containing an epithelial layer with an underlying fibroblast dominated stroma, similar to skin, the cornea provides a physical barrier and protection to the underlying eye tissue [43]. Corneal transparency, required for unimpeded sight, arises largely due to formation of densely packed, regularly spaced, uniform, thin collagen fibrils comprising the well-organized extracellular matrix [10,78]. Following injury however, keratan sulfate proteoglycans, including lumican and keratocan, which are involved in regulation of collagen fibril diameter and spacing, can be downregulated, which can make the cornea “cloudy” and impair sight [10,78]. In a manner similar to skin healing, corneal repair involves mesenchymal and epithelial cell populations, and JNK activation plays a prominent role in regulating cell phenotype. 

Keratocytes, also termed corneal fibroblasts have been demonstrated to activate JNK signaling in response to increasing levels of oxidative stress [79]. Using a corneal alkali burn injury model, activation of stress pathways was observed including activation of JNK [80]. Using *IkB kinase β* conditional knockout mice (IkB binds and inhibits activity *NF*-*κB* [81]), Chen and colleagues demonstrated that phosphorylation and activation of JNK was associated with increased TGFβ1 activity and scarring of the cornea as compared with wild-type mice. In addition, it is known that TGFβ1 activation of JNK1 and JNK2 in keratocytes induces cell migration (unlike TGFβ2), CCN2 expression, downregulation of cornea associated proteoglycans keratocan and lumican, and modulates scar formation by controlling fibronectin and collagen I deposition [10,80,82,83,84,85]. Therefore, JNK signaling in keratocytes in response to injury is a significant regulator of corneal healing and scarring. Interestingly, the inhibition of the TNF-α/JNK pathway using SN50, which inhibits NF-κB signaling, results in increased cell proliferation in mouse corneal wounds [86]. Although performed using epithelial cells, this complements results whereby JNK activation via TGFβ1 suppresses proliferation at the wound edge and inhibits proper healing [87]; TNF-α and TGFβ1 signaling are typically considered antagonistic in their effects on healing [88]. 

With respect to corneal epithelial cells at the wound edge, JNK is activated through mitogen-activated protein kinase kinase kinase 1 (MEKK1), mediating extracellular signal-regulated kinases (ERK), and p38MAP kinase, which induces epithelial cell migration and corneal wound re-epithelialization [89]. This is similar to skin keratinocytes, as discussed in Section 3.1, where p38MAP kinase is also involved in migration as opposed to JNK. JNK activity has also been shown to induce phosphorylation of paxillin on Ser178, formation of focal adhesions and lamellipodia; all necessary events in cellular migration resulting in wound closure [90,91,92]. Additionally, also associated with healing, matrix metalloproteinase-9 (MMP-9) expression is activated through JNK, in combination with Smad3 and NF-κB, via mechanisms involving TAK-1 and MKK4/7, or the TGFβ activated RAS/MEKK1/MKK4 pathway [93]. Therefore, within the cornea, JNK signaling plays a crucial role in epithelial cell migration post injury primarily through modification of cell adhesion and MMP activation [90,91].

Interestingly, apoptosis is hypothesized to be a necessary prerequisite for a normal wound healing response in the cornea [94]. In particular, several different environmental stressors with which JNK activation is associated lead to corneal epithelial cell apoptosis. Lipopolysaccharides (LPS), endotoxins released from Gram-negative bacteria, are known to induce corneal epithelial cell inflammation and apoptosis through phosphorylation of JNK mediated toll-like receptor (TLR) 4 [95]. Apoptosis levels increase further with Rho kinase 1 (ROCK1) overexpression [95]. JNK activation can also be induced by exposure to UV irradiation, a common corneal cell stressor; JNK activation is upstream of apoptosis mediated through p53, polo-like kinase 3 (Plk3), and an efflux of K^+^ [96,97]. Therefore, JNK activation in corneal epithelial cells can be an important regulator of apoptosis prior to migration. 

One of the other primary roles of the cornea is to act as a physical barrier to microbes, with many epithelial layers including the cornea expressing the antimicrobial peptide defensins [98]. The proinflammatory cytokine, IL-1β, induces JNK activation in corneal epithelial cells resulting in activation of NF-κB and subsequent expression of human β-defensins (hBDs) in vitro [99]. Again, stress responses in corneal epithelial cells result in activation of JNK signaling. However, not all stressors result in JNK activation. Indeed, JNK is inhibited in response to membrane leakage, as seen as a result of mechanical wounding [100]. JNK is also not activated when cells are stimulated with hepatocyte growth factor, keratinocyte growth factor, or HSP70, a member of the heat shock protein family known to induce cellular stress [101,102]. The removal or inhibition of transient receptor potential ankyrin 1 (TRPA1), an irritant-sensing ion channel, from mouse corneal fibroblasts is also capable of suppressing JNK activation and results in a reduction of inflammation and scar tissue formation [103]. Therefore, the role of JNK in the cornea is complex and somewhat dependent on the stressor being applied to the cells. 

## 5. Tendon Healing 

Tendon injury and repair in general follows a similar pattern to most soft tissues, inflammation, proliferation, and remodeling [104]. Similar to skin, tendons have a propensity to scar in response to injury [105]. Reducing inflammation post injury has been shown to reduce fibrotic tissue deposition and improve healing outcomes, particularly through the inhibition of TNF-α [106]. Due to their potential to enhance regeneration, tendon stem/progenitor cells (TSCs) have received particular attention with respect to healing. Tarafder et al. (2017) investigated the role of TSCs in regulating inflammation in response to CCN2 [107]. In vivo, using a patellar tendon injury model in Sprague-Dawley rats, they demonstrated that CCN2 promoted an anti-inflammatory response from TSCs. In response to CCN2, the pro-inflammatory markers IL-6, MMP-3, and iNOS positive macrophages were significantly diminished, while expression of anti-inflammatory markers IL-10 and TIMP-3 was significantly upregulated in TSCs as compared with the untreated controls [107]. The authors also reported significant upregulation of IL-10 and TIMP-3 in TSCs in response to CCN2 treatment, as well as significant activation of JNK and STAT3. The importance of MAPK proteins in the anti-inflammatory response observed was confirmed by the addition of the JNK inhibitor, IQ-1s, which negated the CCN2 mediated upregulation of anti-inflammatory markers [107]. A similar study looking at the effects of Aspirin (ASA) on SD rat TSCs, in vivo and in vitro, gave similar findings [105]. The authors observed that ASA treatment significantly reduced IL-6 and MMP-3 expression while upregulating IL-10 and TIMP-3, similar to the effects of CTGF previously observed by Tarafder et al. (2017) [107]. In addition, JNK and STAT3 activity was again significantly upregulated in response to IL-1β mediated inflammation and the anti-inflammatory effects of ASA treatment were diminished by the addition of either a JNK inhibitor, SP6001, or STAT3 inhibitor, S3I201 [105]. Therefore, JNK signaling plays an important role in regulating inflammation in tendon healing. 

The involvement of JNK signaling pathways in tendon healing are now becoming better defined. Skutek et al. (2003) investigated the effects of mechanical stretching, specifically designed to simulate therapeutic stretching, with respect to apoptosis and alteration of JNK expression, using human tendon fibroblasts [108]. After only 15 minutes of mechanical stretching, human tendon fibroblasts increased activation of JNK 1 and JNK 2, as well as an increased rate of remodeling associated apoptosis. Interestingly, this was significantly dampened when cells underwent longer duration (60 min) mechanical stretching [108]. The authors concluded that short duration mechanical stretching induced tendon remodeling associated apoptosis through JNK signaling, but cells developed stress tolerance when exposed to mechanical stretch for longer durations. Overall, it is evident that JNK signaling is prominent in tendon healing, potentially playing multiple roles.

## 6. Oral Tissue Regeneration, Pathology, and Healing

Skin and the oral mucosa are regarded to be two homologous tissues, both characterized by the presence of keratinized epithelium with underlying collagen dense connective tissue. However, there are numerous differences between cutaneous and oral mucosa wounds with respect to healing profiles. Wounds in the oral mucosa heal faster, with minimal scar formation as compared with skin wounds, which is concomitant with reduced inflammatory response, attributed to the reduced recruitment of neutrophils, macrophages, and T-cells [109,110]. Several other contributory factors have been proposed to be involved, such as the presence of saliva, leukocytes, growth factors, phenotypic differences between oral, and cutaneous fibroblasts, as well as the presence of bacteria that stimulate wound healing [109,110]. Unlike skin, in the healing of gingiva, the oral epithelium is highly aggressive; it proliferates and migrates rapidly following injury and can be considered to have a competitive advantage over regeneration of the underlying connective tissue [15]. Despite these differences as is evident in skin, JNK signaling plays important roles in gingival healing. 

### 6.1. Gingival Fibroblasts and JNK 

Fibroblasts constitute the predominant cell type in human gingival connective tissue and play a significant role in tissue remodeling through expression of proteolytic enzymes and synthesis of extracellular matrix glycoproteins and proteoglycans [111]. JNK pathways have been shown to regulate several proteolytic enzymes in human gingival fibroblasts (HGFs) during wound repair. 

Epidermal growth factor (EGF) mediates several of the responses during wound healing and inflammation, such as stimulation of cell proliferation and extracellular matrix turnover, including the expression of MMPs 1, 3, and 13. Urokinase-type plasminogen activator (uPA) is highly expressed in healing gingival tissues [112,113] where it converts plasminogen into plasmin that in turn degrades fibrin and activates matrix metalloproteinases which facilitates tissue remodeling [114]. EGF significantly activates uPA expression in HGFs requiring JNK signaling, and to a lesser extent, the extracellular signal regulated kinases 1/2 signaling pathways [115]. Smith et al. (2006) found that in granulation-tissue fibroblasts from periodontal disease tissues, JNK was activated by TGFβ1 and stimulated uPA production; the JNK inhibitor, SP600125, completely attenuated the stimulus of TGFβ1 on uPA production in a dose-dependent manner [116]. More recently it has been shown that EGF also regulates fibrous tissue remodeling in the periodontal ligament (PDL)-derived cells through both JNK-mediated and MEK/ERK signals by affecting the proliferation, migration, and myofibroblast differentiation [12].

Myofibroblast differentiation has been postulated to be an important cellular transition in gingival wound healing, although we recently reported that α-SMA positive myofibroblasts are absent during gingival healing in a rat model [15] and we have previously shown that myofibroblasts are not associated with drug-induced gingival enlargement (DIGE), a fibrotic condition of the gingiva [117,118,119]. TGFβ1 has been shown to induce myofibroblast differentiation through canonical and noncanonical signaling [120,121,122]. Myofibroblasts are associated with increased cell adhesion and spreading, the maturation of the actin cytoskeleton, and the induction of α-SMA, responses that are regulated by the activity of the RhoA-ROCK and JNK signaling pathways [123]. According to our studies, the exact role of α−SMA associated myofibroblasts in gingival healing appears to have little significance. 

Interestingly, we did recently identify a requirement for JNK signaling in fibronectin and type I collagen synthesis in gingival fibroblasts [124]. More specifically, TGFβ1 has been shown to increase fibronectin synthesis through JNK [124], and it was reported that when human gingival fibroblasts are cultured on periostin coated matrices, a significant increase in fibronectin production was evident, an effect which was attenuated by pharmacological inhibition of either Focal Adhesion Kinase (FAK) or JNK signaling (Figure 4). Interestingly in skin, periostin did not induce matrix synthesis, but instead controlled transition of cells to a myofibroblastic phenotype, a process dependent on β1 integrins and FAK, but not JNK signaling [42]. Moreover, when periostin was added in an exogenous form to diabetic wounds in mice, it similarly induced myofibroblast formation, but not matrix deposition as compared with control collagen scaffolds [40]. 

Providing evidence that FAK and JNK are downstream mediators of periostin induction of fibronectin and type I collagen expression in human gingival fibroblasts, it opens the intriguing possibility that periostin based biomaterials could be used, clinically, as a means of stimulating matrix synthesis during gingival healing [15]. It also provides insights into the pathology of DIGE, an inflammatory condition which we have shown is associated with overexpression of periostin [117,118,119]. Induction of matrix synthesis in the presence of periostin would also partially explain the overgrowth of gingival tissue associated with the condition. The role for JNK signaling in DIGE has not been investigated, although JNK signaling pathways are also involved in the production of inflammatory mediators during gingival inflammation. 

Drug-induced gingival enlargement (DIGE) is a pathological condition that can develop as a side effect due to the systemic administration of the following three types of drugs: anticonvulsants, immunosuppressants and various calcium channel blockers [125]. Both drug-induced and hereditary gingival overgrowth are characterized by the accumulation of extracellular matrix in gingival connective tissues and, particularly, collagenous components, with varying degrees of inflammation. Periostin associated with multiple human fibrotic diseases, is overexpressed in the tissue of gingival overgrowth, as is CCN2, both proteins considered profibrotic. Black et al. (2007) showed that TGFβ1 stimulated CCN2/CTGF levels in HGFs via JNK pathway [126]. Chang et al. (2013) reported that TGFβ1 induced CCN2 expression in buccal mucosa fibroblasts through JNK and p38 MAPK but not ERK signaling [8]. Chen et al. (2012), in their in vitro study, showed that reactive oxygen species, ASK1 and JNK, were involved in the signal transduction of thrombin induced CCN2 expression in HGFs [127]. Thrombin treatment activated JNK, while curcumin inhibited CCN2 expression through JNK suppression. CCN2 is also induced by lysophosphatidic acid (LPA), in oral epithelial cells and gingival fibroblasts [128,129] and this effect can be significantly inhibited by TGFβ type I receptor/ALK5, Smad3, and JNK inhibitors but not ERK, P38, and MAPK inhibitors [129,130]. It has been proposed that LPA produced due to surgical wounds could contribute to the recurrence of gingival overgrowth by upregulating CCN2 expression in HGFs, an effect that is mediated by Smad3 and JNK activation and ALK5 transactivation [130], further highlighting the role of JNK in regulating cellular responses in gingival overgrowth.

### 6.2. Oral Epithelium

As the first barrier against bacterial insults and toxic substances, the junctional epithelium through specialized intercellular junctions, such as E-cadherin junctions, provides a seal between the tooth and the lamina propria of the gingiva [131]. Thus, the structural integrity of the junctional epithelium is of paramount importance for preventing and treating periodontal disease. Evidence suggests that JNK activity is critical to the formation of intercellular E-cadherin junctions between human gingival epithelial cells (hGECs). On natural tooth surfaces, activation of JNK disrupts the intercellular junctions through the dissociation of E-cadherin. The role of JNK in the formation of these E-cadherin junctions was demonstrated by inhibiting JNK pathway which, then, induced the formation of intercellular E-cadherin junctions. Investigating upstream of JNK, activation of the small GTPase RhoA disrupted the formation of E-cadherin junctions between hGECs cells, which was accompanied by JNK activation. Disruption of these intercellular junctions upon RhoA activation was prevented when JNK activity was inhibited. These findings that RhoA plays a role in regulating the intercellular junctions between gingival epithelial cells by activating JNK can be of clinical importance because various bacterial toxins regulate RhoA activity [132]. 

In pathologic conditions, such as in cases where the root surface has been significantly roughened by root planning therapy or the surface of a rough fixture has been exposed due to peri-implantitis or marginal bone loss and apical migration of the junctional epithelium, the epithelial cells interact with rougher surfaces. Recently, it has been shown that on rough substrates with nanometer dimensions, the E-cadherin junctions of hGECs develop slowly or are defective, and that this effect can be reversed by inhibiting JNK [133]. More specifically, JNK inhibition promoted the loss of cortical actin, along with E-cadherin junction development. In contrast, JNK activation induced the development of thick cortical actin, which inhibited both cells spreading and E-cadherin junction development. The dependency on cortical actin in the regulation of the E-cadherin junctions by JNK was further confirmed by showing that cells treated with anisomycin lost the thick cortical actin bundle and developed ECJs when ROCK activity was inhibited to release the tension imposed by cortical actin. All of these results suggest that reducing JNK activity promotes E-cadherin junction development by downregulating the development of the regulatory cortical actin bundle [133]. 

## 7. Dental Pulp 

Dental pulp is a soft, loose connective tissue located in the central portion of each tooth. It is primarily involved in dentin formation and, subsequently, tooth development during childhood and adolescence. Injury to the pulp typically occurs due to dental caries, physical wear, or trauma. In the case of many of the tissues discussed in this review thus far, JNK signaling in fibroblastic cells as a result of injury is associated with cell survival/apoptosis, migration, cellular contractility, and extracellular matrix synthesis. However, dental pulp is unique in that the initial response of the cells present in the pulp is the secretion of tertiary dentin, which can be considered a healing mechanism in order to protect the pulp against mechanical, thermal, and microbial insults [25,26]. 

Similar to inflammation in skin healing, the JNK pathway can participate in the pulp wound repair processes at the initiation stage [134]. The activation of the JNK pathways in the presence of TNF-α has been reported in dental pulp stem cells [135,136], similarly as is evident in dermal fibroblasts. JNKs are also involved in TNF-α signaling regulating the production of pro-inflammatory cytokines such as IL6 and IL8 in human dental pulp stem cells (DPSCs), which is in contrast to neutrophils, where JNK is not involved in pro-inflammatory cytokine production [51]. Interestingly, targeting JNK signaling by its specific inhibitor (SP600125) or resveratrol, significantly attenuated TNFα-induced expression of *IL6* and *IL8* mRNA. Given the negative effects of the persistence of pro-inflammatory cytokines, inhibition of the JNK signaling pathway could be a promising target for therapeutics in the reduction of pulpal inflammation; resveratrol inhibited the phosphorylation of JNK induced by TNF-α in human DPSCs [137]. 

The same research group also demonstrated that JNK and heat-shock proteins are involved in dental pulp cell apoptosis following injury using a cavity preparation in vivo animal model [138]. In normal pulp, c-Jun, but not JNK, was activated in some pulp cells. One-hour post injury, primary apoptosis was evident in odontoblasts, and the arrangement of the odontoblast layer under the cavities was significantly disrupted. Immediately post wounding, p-JNK and p-c-Jun were detected in some pulp cells in the subodontoblastic area, but not in odontoblasts. One day post injury, active JNK and c-Jun were detected in apoptotic pulp cells, whereas HSP70 was detected in nonapoptotic cells [138]. It was also observed that the translocation of HSP70 into nuclei of pulp cells colocalized with active JNK and c-Jun. Four days after injury, active JNK and c-Jun was absent in pulp cells, and HSP70 was present in the cytoplasm but not nuclei. These results suggest that the JNK pathway could be one of the compartments inducing apoptosis in pulp cells, and that HSP70 could have an inhibitory role in the apoptosis of pulp cells during wound healing [138,139].

JNK pathways are not only involved in apoptosis, but are also implicated in neurodegeneration, cell proliferation and differentiation, odontoblast secretory activity, inflammatory responses, and cytokine production by the activation of downstream transcription factors [31,32,33,34,35,36]. JNK signaling pathway have been shown to be responsible for the transduction of WNT4- and WNT6-induced cell migration and differentiation of human dental pulp stem cells [140,141]. Recently, it was also shown that WNT7B can promote the migration and differentiation of human dental pulp cells through the WNT/β-catenin and JNK signaling pathways [134]. JNK also partially regulates the inflammatory response of DPSCs downstream of WNT5A [142]. 

One of the central cells important in pulp biology is the odontoblast and it is known that *c-Jun* and *jun-B* genes can play roles in surviving odontoblasts and odontoblast-like cells [139]. During the early phase of reparative dentinogenesis, levels of c-Jun and jun-B increased in dental pulp cells within and around the reparative dentin matrix formed adjacent to the cavity floor [139]. At later stages of wound healing post cavity preparation, c-Jun and jun-B were expressed only in pulp cells lining the irregular surface of the thick reparative dentin. These results suggest that c-Jun and jun-B can play important roles in tertiary dentin formation [139]. The same research group also demonstrated that JNK and heat-shock proteins are involved in dental pulp cell apoptosis following injury using a cavity preparation in vivo animal model [138]. In normal pulp, c-Jun, but not JNK, was sparsely activated in pulp cells, but one-hour post injury, primary apoptosis was evident in odontoblasts, and the arrangement of the odontoblast layer under the cavities was significantly disrupted. Immediately post wounding, p-JNK and p-c-Jun were detected in some pulp cells in the subodontoblastic area, but not in odontoblasts. One day post injury, active JNK and c-Jun were detected in apoptotic pulp cells, whereas HSP70 was detected in nonapoptotic cells [138]. It was also observed that the translocation of HSP70 into nuclei of pulp cells colocalized with active JNK and c-Jun. Four days after injury, active JNK and c-Jun were absent in pulp cells, and HSP70 was present in the cytoplasm but not nuclei. These results suggest that the JNK pathway could be one of the compartments inducing apoptosis in pulp cells, and that HSP70 could have an inhibitory role in the apoptosis of pulp cells during wound healing [138,139].

## 8. Conclusions

First described in 1990 by Kyriakis and Avruch [143], JNKs are well described to be activated in response to cellular stress, but novel roles for this intracellular signaling cascade continue to be defined (Figure 2). As is evident from studies in *Drosophila*, skin, cornea, and gingiva, JNKs play an important role in epithelial cell biology in response to injury, regulating migration, differentiation, as well as apoptosis. JNKs play equally important roles in fibroblast behavior, regulating transition to the myofibroblast phenotype, as well as expression of inflammatory mediators and matrix degrading enzymes. In oral tissues, JNKs are implicated in the regulation of gingival fibroblasts, PDLSCs, odontoblasts, and oral epithelium to inflammatory mediators and profibrotic cytokines (Figure 5). Our recent finding that JNK signaling is required for fibronectin upregulation in gingival fibroblasts in response to the matricellular protein provides intriguing new roles for JNKs in the control of matrix synthesis in oral healing [15]. Future research would likely find new roles for JNKs in the regulation of cell behavior beyond the defined roles in cellular stress. 

## Figures and Tables

**Figure 1 ijms-21-01015-f001:**
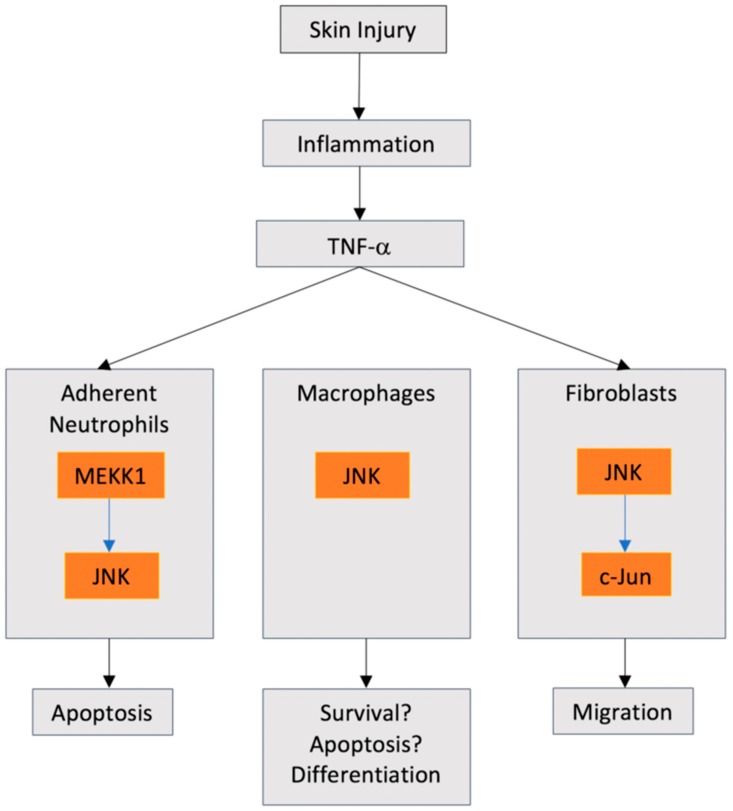
Possible interplay of c-Jun N-terminal kinases (JNK) signaling in the inflammatory stage of skin healing. Production of tumor necrosis factor α (TNF-α ) has been demonstrated to have defined roles in neutrophils and fibroblasts through JNK signaling, with the role of JNK signaling in macrophage biology in response to TNF-α less well defined in healing specifically.

**Figure 2 ijms-21-01015-f002:**
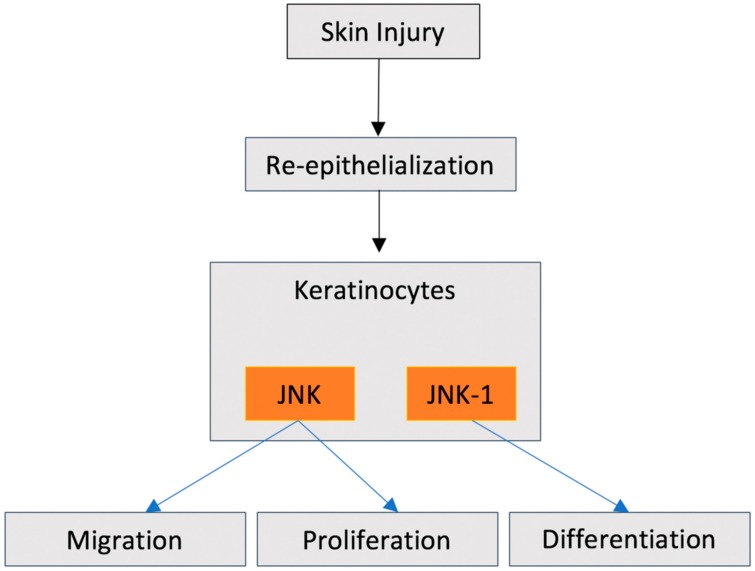
Contrasting roles of JNK signaling in epithelial biology post wounding in skin. JNK-1 has been shown to influence terminal differentiation of keratinocytes, but similarly JNK has been shown to play a role in proliferation and migration.

**Figure 3 ijms-21-01015-f003:**
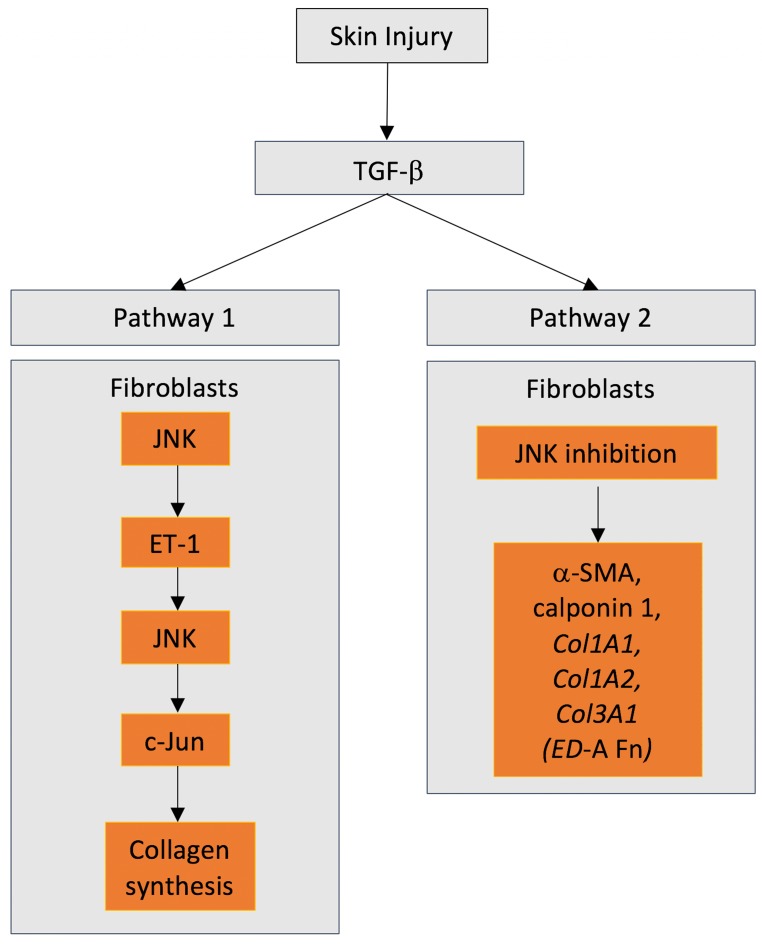
Conflicting role of JNKs in dermal fibroblast biology. JNK activation (Pathway 1) and JNK inhibition (Pathway 2) have been described to induce differentiation of fibroblast towards a contractile, matrix secreting myofibroblast phenotype.

**Figure 4 ijms-21-01015-f004:**
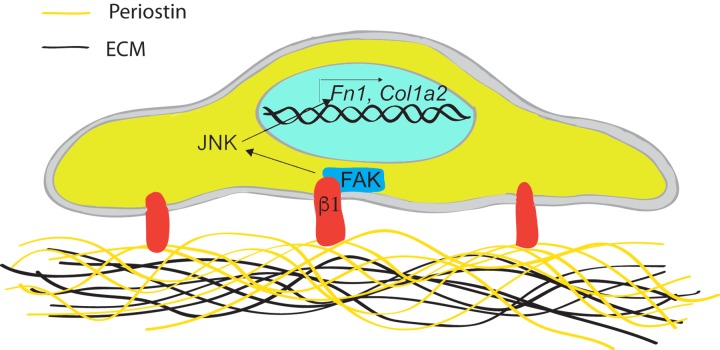
Periostin induces fibronectin and *Col1a2* in human gingival fibroblasts in a manner dependent on JNK signaling. Requiring β1 integrin engagement into adhesion sites and FAK phosphorylation, JNK activation is required to induce a matrix secreting phenotype.

**Figure 5 ijms-21-01015-f005:**
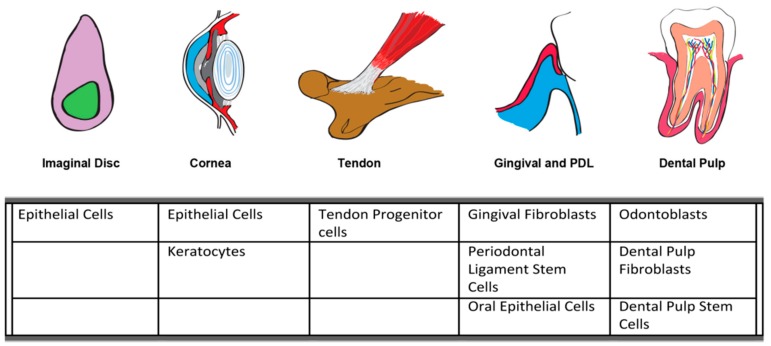
Summary of cell types in which JNK signaling plays a role in soft connective tissues.

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
