# Peer review of "JNK Signaling as a Key Modulator of Soft Connective Tissue Physiology, Pathology, and Healing"

_ijms, 2020, doi:10.3390/ijms21031015_

Round 1
Reviewer 1 Report
This review paper was written very well and complihensive. One figure which depict upstream and downstream cascades of JNK enzyme regarding connective tissue wound repair will be helpful.
Author Response
Reviewer 1
This review paper was written very well and comprehensive. One figure which depict upstream and downstream cascades of JNK enzyme regarding connective tissue wound repair will be helpful.
We thank the reviewer for their comments and agree that a figure is required. We have added a figure showing the interplay of JNK activity and the regulation of events post wounding in connective tissue. We have largely focused it on skin as several conflicting results exist on the role of JNK signaling between tissue types.
Reviewer 2 Report
The manuscript titled “JNK Signaling as a Key Modulator of Soft Connective Tissue Physiology, Pathology and Healing” reviewed the role of c-Jun N-terminal Kinases (JNK) in the regulation of cellular activities in drosophila, cornea, tendon, and oral tissues. However, the whole review is not well balanced as more than half of the efforts were put into the author’s preferred field of oral tissues. Further, it’s difficult for the reader to follow as the contents were not well organized. Last but not least, the majority of the references are more than 5-years old that may not reflect the most recent update of the fields.
A few comments for future improvement:
Line 37, “expresses a specific phenotype” should be replaced. Line 95-96, please re-write the sentence. Line 303—5, use JNK instead of “c-Jun N-terminal Kinases”. Line 435, “would healing” should be “wound healing”. Line 487, “expression of p-focal adhesion kinase, p-FGFR” should be changed as kinase phosphorylation is a modification instead of expression. Provide better figures with figure legends.
Author Response
The whole review is not well balanced as more than half of the efforts were put into the author’s preferred field of oral tissues.
We thank the reviewer for their honest opinion of the review. Upon reflection we agree with the reviewer that there was an imbalance between topics. We have added a section on skin healing in which much of the work has been done on JNK signaling and wound repair. We have also reduced the size of the section on the oral tissues.
Further, it’s difficult for the reader to follow as the contents were not well organized. Last but not least, the majority of the references are more than 5-years old that may not reflect the most recent update of the fields.
We have significantly altered the readability of the manuscript and agree it did not flow particularly well, especially between sections. We have used the most-updated references related to JNK in the tissues we have focused on, but much of the work for certain tissues was not performed within the last 5 years.
Line 37, “expresses a specific phenotype” should be replaced. Line 95-96, please re-write the sentence. Line 303—5, use JNK instead of “c-Jun N-terminal Kinases”. Line 435, “would healing” should be “wound healing”. Line 487, “expression of p-focal adhesion kinase, p-FGFR” should be changed as kinase phosphorylation is a modification instead of expression. Provide better figures with figure legends.
The manuscript has been proofread and all changes made. We have also added additional figures and improved figure legends.
Round 2
Reviewer 2 Report
The overall quality of the manuscript was significantly improved after including skin healing, in which JNKs signaling plays a substantial role. The authors also improved the readability of the manuscript, depicting major events with flow charts. I would recommend “Accept after minor revision”.
Comments:
Line 41-42, “architecture and soluble signals” may change back to original words to avoid misleading. Figure 3, please remove the red underline with “EC-A Fn”. Figure 5, the words in the table are blurred at the high magnitude, please replace the table with a version with higher quality (DPI > 300).
Author Response
We thank the reviewer for their comments. We have addressed all as requested.